# Usefulness of Atherogenic Indices for Predicting High Values of Avoidable Lost Life Years Heart Age in 139,634 Spanish Workers

**DOI:** 10.3390/diagnostics14212388

**Published:** 2024-10-26

**Authors:** Tomás Sastre-Alzamora, Pedro J. Tárraga López, Ángel Arturo López-González, Daniela Vallejos, Hernán Paublini, José Ignacio Ramírez Manent

**Affiliations:** 1Research ADEMA SALUD Group, University Institute for Research in Health Sciences (IUNICS), 07010 Palma, Spain; tsastre04@sonrie.com (T.S.-A.); d.vallejos@eua.edu.es (D.V.); h.paublini@eua.edu.es (H.P.); joseignacio.ramirez@ibsalut.es (J.I.R.M.); 2Faculty of Medicine, UCLM (University of Castilla La Mancha), 02008 Albacete, Spain; pjtarraga@sescam.jccm.es; 3SESCAM (Health Service of Castilla La Mancha), 02008 Albacete, Spain; 4Faculty of Dentistry, ADEMA University School, 07010 Palma, Spain; 5Institut d’Investigació Sanitària de les Illes Balears (IDISBA), Health Research Institute of the Balearic Islands, 07010 Palma, Spain; 6Health Service of the Balearic Islands, 07010 Palma, Spain; 7Faculty of Medicine, University of the Balearic Islands, 07010 Palma, Spain

**Keywords:** cardiovascular diseases, atherogenic indices, heart age, preventive activities, lifestyle habits

## Abstract

Background: Cardiovascular diseases (CVDs) are the leading cause of morbidity and mortality worldwide, accounting for one-third of all global deaths. The World Health Organization (WHO) asserts that prevention is the most effective strategy to combat CVD, emphasizing the need for non-invasive, low-cost tools to identify individuals at high risk of CVD. Atherogenic indices and heart age (HA) are valuable tools for assessing cardiovascular risk (CVR). The aim of our study was to evaluate the association between atherogenic indices and HA. Methods: A cross-sectional study was conducted involving 139,634 Spanish workers to determine the association between three atherogenic indices and HA. ROC curves were employed to identify the cut-off values for the various atherogenic indices used to estimate high HA. The cut-off points, along with their sensitivity, specificity, and Youden index, were determined, and the area under the curve (AUC) was calculated. Results: As the values of the atherogenic indices increased, so did the risk of having elevated avoidable lost life years (ALLY) HA. In the ROC curve analysis, the AUC with the best results corresponded to the total cholesterol/HDL-c atherogenic index, with an AUC of 0.803 in females and 0.790 in males. The LDL-c/HDL-c atherogenic index showed an AUC of 0.780 in women and 0.750 in men, with Youden indices around 0.4. When analyzing the AUC of the atherogenic index for triglycerides/HDL-c, the results were 0.760 in women and 0.746 in men. Conclusions: Atherogenic indices and HA show a close relationship, with an increase in these indices leading to a rise in HA values. Raising patient awareness that as their CVR levels increase, so does their HA may be useful in achieving some benefit in reducing CVR.

## 1. Introduction

Cardiovascular diseases (CVDs) are the leading cause of morbidity and mortality worldwide [1], accounting for approximately one-third of all global deaths [2,3], with about three-quarters of these deaths occurring in low- and middle-income countries. In this context, the identification and management of cardiovascular risk factors (CVRFs) are crucial for the prevention and treatment of these diseases [4]. Traditional risk factors include hypertension [5], dyslipidaemia [6], diabetes mellitus [7], smoking [8], physical inactivity [9], and obesity [10]. However, in recent decades, new markers and tools have been developed and validated to more accurately assess individual cardiovascular risk (CVR). Among these, atherogenic indices [11,12] and the concept of “heart age” [13,14] have gained prominence as complementary indicators in cardiovascular risk assessment.

Atherogenic indices are biochemical parameters that reflect the relationship between different lipid fractions in the blood and are used to assess the risk of atherosclerosis [15,16] and cardiovascular events [17,18]. The most common include the total cholesterol/HDL-c index [19], the atherogenic index of plasma (AIP) [20,21], and the non-HDL/HDL cholesterol ratio [22]. These indices provide a more integrated measure of lipid profiles compared to the traditional measurements of total cholesterol, LDL, and HDL separately [17].

The total cholesterol/HDL-c index is one of the oldest and most widely used atherogenic indices. An elevated TC/HDL ratio has been shown to be associated with a greater risk of cardiovascular events [23]. A high value indicates a predominance of atherogenic particles in the blood, suggesting greater potential for the formation of atheromatous plaques in the arteries [24].

The AIP is calculated as the ratio between triglycerides and HDL cholesterol (TG/HDL-C). This index has been proposed as a marker for the size and number of LDL particles, with higher values being associated with smaller and denser LDL particles, which are more atherogenic [25]. Several studies have demonstrated that AIP is an independent predictor of cardiovascular disease and can be particularly useful in assessing risk in patients with metabolic syndrome [26,27] and type 2 diabetes mellitus [28,29].

The concept of “heart age” (HA) refers to a CVR assessment tool that translates an individual’s absolute risk into an equivalent age of their cardiovascular system [30]. This approach aims to communicate risk in a more understandable and motivating way for patients [31]. HA is calculated using algorithms based on traditional risk factors such as age, sex, blood pressure, cholesterol levels, the presence of diabetes, and smoking habits [32].

Several online calculators and mobile applications are available for healthcare professionals and patients to calculate HA [33]. However, interpreting the results can be challenging, especially for patients, as entering the same data can yield different outcomes [34]. One of the most commonly used algorithms is the one developed by the Framingham Heart Study [35], which estimates HA based on the Framingham Risk Score. This tool not only helps estimate CVR, but can also be used to monitor the impact of therapeutic interventions and lifestyle changes.

HA is particularly useful in clinical practice because it translates abstract risk data into a more tangible message. For instance, informing a 40-year-old patient that they have a HA of 60 can serve as a strong motivator for adopting lifestyle changes or adhering to medical treatments. Some studies have shown that communicating CVR in terms of HA can improve adherence to lifestyle and therapeutic recommendations, ultimately reducing the risk of cardiovascular events [36,37]. In recent decades, the World Health Organization (WHO) has emphasized that the best strategy to prevent CVDs is prevention [38], making it crucial to have non-invasive, low-cost tools to identify individuals at high risk of developing CVDs.

The integration of atherogenic indices and HA into clinical practice can provide a more comprehensive and accurate assessment of CVR. While atherogenic indices offer detailed insights into lipid profile and atherosclerosis risk, HA translates overall CVR into a format that is easily understandable and motivating.

To effectively implement these approaches in clinical practice, a multifaceted strategy is recommended, encompassing patient education, the use of digital tools, and the integration of these parameters into electronic medical record systems. Patient education is crucial for understanding the importance of these indices and the relevance of heart age. Employing mobile applications and online calculators can facilitate access to and the use of these tools for both healthcare professionals and patients.

The benefits of using atherogenic indices and HA include more accurate risk assessment, improved risk communication to patients, and increased motivation to adopt lifestyle changes and adhere to therapies. However, challenges in implementation include the need for education and training for healthcare professionals, integration into existing health systems, and variability in available calculators and algorithms.

The aim of our study was to assess whether there is an association between atherogenic indices and HA in order to influence patients’ engagement in preventive activities, as recommended by the WHO, and to encourage patients to modify harmful lifestyle habits.

If we find an association between atherogenic indices and heart age, we believe that one of the most effective ways to motivate patients to change harmful lifestyle habits is by making them aware of the potential years of life lost. The clinical utility of this study lies in the fact that by calculating the patient’s atherogenic index during consultation, we can make them aware of the potential years of life lost, thereby emphasizing the urgency of adopting healthier lifestyle habits.

## 2. Materials and Methods

### 2.1. Participants

A total of 139,634 Spanish workers from various areas and labor sectors (mainly public administration, health, hospitality, construction, and commerce) participated in a descriptive, cross-sectional survey. Employees who underwent occupational medicine exams between January 2019 and June 2020 were chosen to be part of the study. Examine the flowchart in Figure 1.

The inclusion criteria to select the sample are detailed below:Individuals aged 20 to 69 years old.Consent to take part in the research.Permission given for the data to be used for epidemiological research.Being employed by one of the companies participating in the research and not being temporarily disabled at the time of the study.

### 2.2. Determination of Variables

Following process standardization to prevent interobserver bias, occupational health professionals from the participating firms conducted all measurements, whether anthropometric (height, weight, and waist circumference), analytical, or clinical.

#### 2.2.1. Anthropometric Determinations

Measurements of height (in cm) and weight (in kg) were taken using a SECA 700 scale with a stadia meter. Measurements were taken with the person wearing underwear according to the international standards for anthropometric evaluation of the ISAK [39].

When the subject was standing straight, feet together and abdomen relaxed, waist circumference was measured using a tape measure parallel to the floor at the level of the last floating rib.

#### 2.2.2. Clinical Determinations

Blood pressure was measured after 10 min of rest, with the subject seated. Three measurements were made at one-minute intervals, and the average of the three was calculated.

#### 2.2.3. Analytical Determinations

The blood sample was taken after a minimum of 12 h of fasting, and processed in 48 to 72 h. Measurements of triglycerides, total cholesterol, and blood sugar were performed automatically by enzymatic procedures. The Cl2Mg dextran sulphate precipitation process was employed for HDL-cholesterol.

By using the Friedewald formula, which is only reliable when triglycerides do not exceed 400, LDL-cholesterol can be calculated indirectly. Each and every analytical parameter is stated in mg/dL [40].

#### 2.2.4. Risk Scales

The following formulae were used to calculate the different atherogenic indices:Total cholesterol/HDL-c [19].LDL-c/HDL-c [41].Triglycerides/HDL-c [42].

Various cut-off thresholds were determined for each index based on data already available in the literature.

Total cholesterol/HDL-c was categorized as high risk if it was above 7% or 9%; moderate if it was between 4.5 and 7% or 5 and 9%; and low risk below 4.5% or 5% for women and men, respectively. The LDL-c/HDL-c index risk was low if below 3% and high above 3%. The triglyceride/HDL ratio was considered to be elevated at 3% and above [43].

HA values were found using the “Heart Age Calculator” feature on the website http://www.heartage.me, accessed on 7 July 2020. We have verified that the website is no longer available. The formula used to calculate cardiac age included total cholesterol, HDL cholesterol, fasting glucose, smoking, BMI, and systolic and diastolic blood pressure, with scores similar to the original formula. Based on these values, a series of points were added or subtracted from the chronological age, which were also adjusted by sex. The final result corresponded to the individual’s heart age. In Table 1, we provide the values assigned to each of the variables used to calculate HA. The scale can be used with people who are between 18 and 80 years old. The highest number of years that can be gained or maintained compared to chronological age (CA) is 20, while the maximum age cut-off for HA is 19 years [44]. Avoidable lost life years (ALLY) [45] are defined as the difference between CA and HA. CA and HA are allies. This concept also addresses heart aging, with 17 as the cut-off age for high ALLY heart age [42].

Anyone who had smoked at least one cigarette in the previous thirty days (or its equivalent in other forms of consumption) or who had stopped smoking less than a year before was considered a smoker.

The 2011 national classification of occupations (CNAE) and the Spanish Society of Epidemiology’s criteria were used to divide social class into three groups: class I, which included directors/managers, university professionals, athletes, and artists; class II, which included intermediate occupations and self-employed workers without employees; and class III, which included unskilled workers. This was carried out in compliance with the suggestion made by the Spanish Society of Epidemiology’s social determinants committee [46].

There were three designated categories for educational attainment: elementary, high school, and university.

Strong adherence was defined as results equal to or higher than 9 in the 14-question Mediterranean diet adherence questionnaire [47], which has a score range of 0 to 1 for each question. Amount of physical activity was measured using the International Physical Activity Questionnaire, or IPAQ [48]. Alcohol consumption was measured in alcohol units (AUs)—in Spain, one AU is equivalent to 10 g of pure ethanol. It is considered high when a woman consumes 14 AUs and a man 21 AUs each week [49].

### 2.3. Statistical Analysis

For each of the categorical variables, a descriptive analysis was carried out to determine the frequency and distribution of the responses. The mean and standard deviation for quantitative data were computed using a normal distribution. Student’s t test for independent samples (for mean comparison) and the chi^2^ test (with Fisher’s exact statistic correction when necessary) were used for bivariate association analysis. The variables linked to the most important risk factors were identified using multivariate approaches. Multivariate analysis was conducted using logistic regression, and Hosmer–Lemeshow goodness-of-fit and odds ratios were computed. ROC curves were used to determine the cut-off values for the various atherogenic indices used to estimate high cardiac age. The cut-off points with their sensitivity, specificity, and Youden index were determined, along with the area under the curve (AUC). The Statistical Package for the Social Sciences (SPSS) version 28.0 for Windows was used to conduct the statistical analysis, with an accepted statistical significance level of 0.05.

## 3. Results

The anthropometric and clinical details of the study participants are displayed in Table 2. The analyses comprised 139,634 individuals, of whom 83,282 were men (59.6%) and 56,352 were women (40.4%). The average age was just over 40, with the majority of the group being in the 30-to-49-year-old age range. Men had more negative anthropometric, clinical, and analytical findings. With only an elementary education, the majority of the labor force belonged to social class III. Most men did not follow a balanced diet or engage in regular physical activity (the situation was better for women). Approximately one-third smoked.

The mean values, in both sexes, of ALLY heart age increased in parallel with the increase in values of the different atherogenic indices. In all cases, ALLY HA values were higher in men. The differences observed in all cases showed statistical significance (see Table 3).

Similarly, when stratifying by age, the mean values of ALLY heart age increased in both sexes as age increased. The same trend is observed in the values of the different atherogenic indices (Table 4).

Something similar was observed when assessing the prevalence of high ALLY HA values, i.e., the prevalence increased as the values of all atherogenic indices increased. Prevalence was always higher in men. In all cases, as can be seen in Table 5, the differences observed were statistically significant.

Table 6 shows the results of the multinomial logistic regression analysis stratified by age groups. Sex, social class, educational level, tobacco and alcohol consumption, physical activity, adherence to the Mediterranean diet, and levels of various atherogenic indices have been established as independent variables.

It is observed that the effect of the different independent variables on the appearance of elevated heart age values is not always the same across the different age groups.

Sex, social class, and educational level behave similarly in all groups. Tobacco consumption and the various atherogenic indices show higher odds ratios in younger people, with their influence decreasing as a person’s age increases. On the other hand, alcohol consumption, physical inactivity, and low adherence to the Mediterranean diet increase their influence (odds ratio values) as a person ages.

In Figure 2 and Table 7, we see how the areas under the curve of all atherogenic indices for predicting the presence of high ALLY HA are moderate–high, especially in women.

Table 8 displays the cut-off points, along with their sensitivity, specificity, and Youden index, for the various atherogenic indices in predicting high ALLY heart age (HA). In all cases, Youden indices are higher in women.

## 4. Discussion

Cardiovascular diseases remain the leading cause of death worldwide and have been increasing in recent years. Atherosclerosis is the primary pathophysiological factor underlying these diseases. Another CVRF is so-called HA, which estimates cardiac aging. The rise in CVDs is significantly influenced by population aging, harmful lifestyle habits, urban living conditions, and obesity [50]. In recent decades, the WHO has emphasized that preventive measures are the most effective strategy to reduce cardiovascular morbidity and mortality. Therefore, the early identification of CVRF and the implementation of motivating interventions are essential to encourage patients to adopt healthy lifestyle habits [51].

Changes in diet within modern society—particularly in large cities—such as the rapid consumption of food due to time constraints and excessive reliance on prepared meals, processed products and snacks (including popcorn, potato chips, salted crackers, sunflower seeds, desserts, sweets, sugary or carbonated drinks, etc.), coupled with a lack of physical exercise and increased screen time (computer, mobile phones, television, video games, etc.), along with the rising rates of obesity [52] driven by these factors, contribute to lipid disorders that create the conditions necessary for the formation of atherosclerotic plaques in arteries [53]. In our sample, 65.8% of men and 52.8% of women did not follow a Mediterranean diet. A similar pattern was observed with regular physical activity, where 48.6% of women adhered to it, compared to only 37.6% of men. In the multinomial logistic regression analysis stratified by age groups in our study, the influence of these two variables increases as the population ages. Physical activity exerts a significantly greater influence (OR) than the Mediterranean diet on high ALLY HA.

Among harmful habits, smoking is one of the primary risk factors for developing atherosclerosis, as it exacerbates dyslipidaemias [54]. Smokers have a three to six times higher risk of developing atherosclerosis compared to non-smokers, and they typically develop it ten years earlier [55,56]. In our study, 33.2% of men and 32.1% of women were smokers despite anti-smoking campaigns in our country and public health education programs. In the multinomial logistic regression analysis, we observe that as age increases, smoking shows lower ORs for high ALLY HA, indicating a reduced influence in older populations. This highlights the critical importance of early intervention to mitigate the impact of smoking on HA.

Excessive alcohol consumption has also been shown to be detrimental to cardiovascular health. One challenge is defining what constitutes excessive alcohol consumption, as there is no international consensus on this definition. Various studies have described the effects of alcohol on the cardiovascular system as a J-shaped curve, where moderate drinkers might have a lower CVR compared to abstainers, while heavy drinkers have a higher CVR [57,58]. However, there is insufficient evidence to recommend reducing alcohol consumption as a preventive measure for CVDs [59,60]. In this variable, our population showed that the percentage of men who consumed alcohol regularly was twice that of women, with 32.7% of men compared to 15.6% of women. In our study, we found that alcohol consumption was detrimental across all age groups, with a 343% increased risk in the youngest cohort (20–29 years), progressively worsening across older age groups. As the individual’s age increases, alcohol exerts a greater influence on high ALLY HA, with the OR rising from 3.43 in the youngest group to more than double at 7.45 in the oldest age group (60–70 years).

These lifestyle habits, already proven as CVRFs, show worse outcomes in men than in women, as reflected in all of our results. Both the mean values of ALLY heart age according to levels of different atherogenic indices by sex and the prevalence of high heart age values according to levels of different atherogenic indices by sex consistently demonstrate worse outcomes for men compared to women. Among the variables influencing healthy lifestyle habits, social class and education level have been described as key factors [61]. Our results highlight the impact of education level on high ALLY heart age, where we observe a consistent influence across all age groups studied. Individuals with only elementary school education show an OR ranging between 2.4 and 2.68 for high ALLY HA, underscoring the role of education in cardiovascular health.

Low socioeconomic status—which comprises factors such as educational attainment, income level, stable employment, and access to housing [62]—has also been associated with increased cardiovascular morbidity and mortality. Increased CVR in individuals from lower socioeconomic backgrounds has been linked to chronic stress, as well as a higher prevalence of other CVRFs (hypertension, obesity, smoking, dyslipidemia, diabetes), which are often associated with lower educational levels and limited purchasing power [63,64]. These findings are consistent with the data obtained in our study, where social class similarly influences all age groups analyzed. We observe that individuals in social class III have an OR more than double that of those in social class I. Additionally, this variable worsens with age; for instance, the OR in social class III increases from 2.41 in the 20–29 age group to 5.21 in the oldest group (60–70 years). A similar pattern is seen between social classes I and II, where social class II consistently shows higher ORs across all age groups. Interestingly, in social class II, the highest ORs are observed in the youngest population group, with differences ranging from 4% to 14% compared to the other age groups.

All these CVRFs lead to changes in an individual’s metabolism that facilitate the development of prediabetes, insulin resistance, diabetes, dyslipidemia, obesity, and metabolic syndrome [65,66,67]. These conditions themselves are also risk factors for CVDs and contribute to the progression of atherosclerosis and increased cardiovascular mortality [68,69,70,71].

The formation and progression of atherosclerotic plaques are strongly related to the presence of CVRFs [72]. Failure to intervene early and in a multidisciplinary manner on these risk factors allows for the progression of atherosclerotic lesions over time and leads to complications that increase mortality [73].

Among the latest tools developed to assess CVR are atherogenic indices, which evaluate the risk of atherosclerosis at a low cost and are non-invasive, unlike molecular biomarkers, which, although highly specific, are expensive and therefore not efficient for the general population [74].

In our study, we evaluated a sample of 139,634 individuals from different autonomous communities in Spain. We evaluated 83,282 men and 56,352 women, making it a representative sample of the Spanish population. Most participants had attained only an elementary education and belonged to social class III. The mean age of the sample was slightly over 40 years, with around 60% within the age range of 30 to 49 years. Anthropometric, clinical, and analytical parameters revealed worse results in men. Among unhealthy lifestyle habits, it is noteworthy that one-third of the sample were smokers and two-thirds of the men did not engage in regular physical activity or follow a Mediterranean diet, compared to approximately 50% of the women.

When studying the association between atherogenic indices and ALLY HA, we found that the latter increased as the mean values of atherogenic indices rose. A similar pattern was observed when evaluating the prevalence of elevated ALLY HA values; that is, prevalence increased as the values of all atherogenic indices increased. Prevalence was consistently higher in men. In all cases, the observed differences were statistically significant.

Multinomial logistic regression analysis revealed that both sex and higher values of the different atherogenic indices increase the risk of having elevated ALLY HA values. The significant elevation in the odds ratio indicates a strong association between exposure and the event. These results reinforce the findings of other studies that have linked atherogenic indices with cardiovascular mortality [75,76].

In the ROC curve analysis for the three atherogenic risk formulas evaluated, the areas under the curve (AUC) with the best results correspond to the total cholesterol/HDL-c index, with a value of 0.803 in females and 0.790 in males, and the LDL-c/HDL-c atherogenic index, with a value of 0.780 in females and 0.750 in males. Both are above 0.75, indicating good performance. However, the Youden indices are not excessively high, around 0.4.

Upon analyzing the AUC of the triglycerides/HDL-C atherogenic index, the result was found to be 0.760 in females (considered good) and 0.746 in males, which is close to 0.75 but does not quite reach this value. This suggests that, strictly speaking, the test result should be regarded as moderate. The Youden index was lower in both cases, indicating lower sensitivity and specificity.

We found no studies in the literature that evaluate this specific association, so we cannot compare our results with those obtained by other authors. To address this gap, we attempt to compare our findings with articles assessing the relationship between atherogenic indices and cardiovascular risk.

Clearly, another risk factor for CVDs is an aging population [77], which constitutes a non-modifiable risk factor. Therefore, our efforts should focus on promoting healthy lifestyle habits that enhance quality of life and facilitate healthy aging.

The study by Hernáez et al. (2019) highlights that while aging has been associated with lower LDL cholesterol levels in various studies, CVR increases with age as LDL cholesterol is highly atherogenic [78]. This could correlate with our findings, where an increase in atherogenic risk corresponds with a rise in HA. Additionally, a study involving 178,282 working women examined various CVRFs and atherogenic indices, observing in a multivariate analysis that age was the only factor affecting all the CVR scales analyzed [79].

Other studies evaluating cardiovascular risk using various scales such as REGICOR, SCORE, and atherogenic indices also found an increase in CVR with age, despite this not being the primary objective of the studies [80,81,82].

Our findings show that as age increases, so do the three atherogenic indices evaluated. The same pattern is observed with ALLY HA, with the mean standard deviation increasing nearly tenfold between the first decade (20–29 years) and the last decade (60–70 years) analyzed. Notably, across all age groups, the ALLY HA values are lower in women than in men, which could be influenced by the less atherogenic lipid profile observed in women in our study. This aligns with the publication by Fappi et al. (2020), which details the relationship between cardiometabolic profile and chronological age [83]^.^

Although chronological heart age is undeniably associated with an individual’s CA, it is clear that various factors influence myocardial cells at the molecular and cellular level. Therefore, the heart grows and ages under the influence of these factors. This means that the biological age of the heart may not align with CA. Consequently, the biological age of the heart could be greater than the individual’s CA, potentially increasing cardiovascular morbidity and mortality [84].

As previously discussed, the best measure to reduce cardiovascular mortality is prevention. To achieve this, it is essential to be able to assess CVR easily and have sufficiently motivating and incentivizing tools to encourage patients to change their unhealthy lifestyle habits. This way, patients can understand their risk in a straightforward manner and learn how their actions can reduce it. Several studies have already demonstrated the effectiveness of different tools in motivating patients, such as the concept of “pulmonary age” for smoking cessation [85]; “vascular age” to reduce cardiovascular risk factors [86]; and “heart age” via electrocardiogram to lower cardiovascular risk factors [87].

One meta-analysis published in January 2024, examining the effect of communicating CVR to patients, its influence on their risk perception, and changes in their behaviors and risk factors, concluded that informing patients about their CVR has a positive impact on risk awareness and the intention to change risk factors. Patients experienced a reduction in overall risk scores between 6 and 12 months of follow-up, along with improvements in blood pressure and cholesterol levels. These results were more pronounced in patients with no established CVDs. There was a significant shift towards a healthier diet and increased motivation for initiating and adhering to necessary medication. The authors concluded that effectively informing patients about their cardiovascular risk levels can lead to meaningful reductions in overall vascular risk [88].

According to the results of our study, there is a close relationship between atherogenic indices, which assess the risk of atherosclerosis, and HA, which evaluates cardiac aging. As the values of these indices increase, there is also an increase in ALLY HA.

The ability to communicate a patient’s cardiovascular risk is of paramount importance, particularly in public health and preventive activities. We believe that explaining to patients that as their atherogenic index increases, so does their chances of experiencing a cardiovascular event, along with the aging of their heart, may serve as a more graphic and comprehensible method. This approach could be highly beneficial in clinical practice, acting as a motivating tool to encourage lifestyle changes and reduce the CVR of our patients.

## 5. Strengths and Limitations

The strengths of this study include the large sample size—which exceeds 139,000 individuals—and the lack of existing studies addressing a similar topic, making this study a potential reference for future research.

The main limitation is that HA is assessed through validated risk calculators rather than objective methods, due to the impracticality of performing objective measurements in such a large sample.

Another significant limitation of our study is the lack of records for patients undergoing treatment with lipid-lowering medications, which means that some results could potentially be underestimated.

Further, our data were derived from routine occupational health screenings among the working population in Spain aged 20 to 69 years. Consequently, the findings of this study may not be generalizable to other populations with different demographic characteristics.

## 6. Conclusions

Atherogenic indices and HA show a close relationship, with increases in these indices being reflected by a rise in HA values.

The effect of different dependent variables on the occurrence of elevated HA values is not consistent across all age groups.

Sex, social class, and educational level behave similarly across all age groups.

Smoking and the studied atherogenic indices show higher odds ratios in younger individuals, with their influence decreasing as the person’s age increases.

Alcohol intake, physical inactivity, and low adherence to the Mediterranean diet increase their influence as the individual ages.

Raising patients’ awareness that their HA increases as their cardiovascular risk levels rise could be useful in achieving benefits in cardiovascular risk reduction.

## Figures and Tables

**Figure 1 diagnostics-14-02388-f001:**
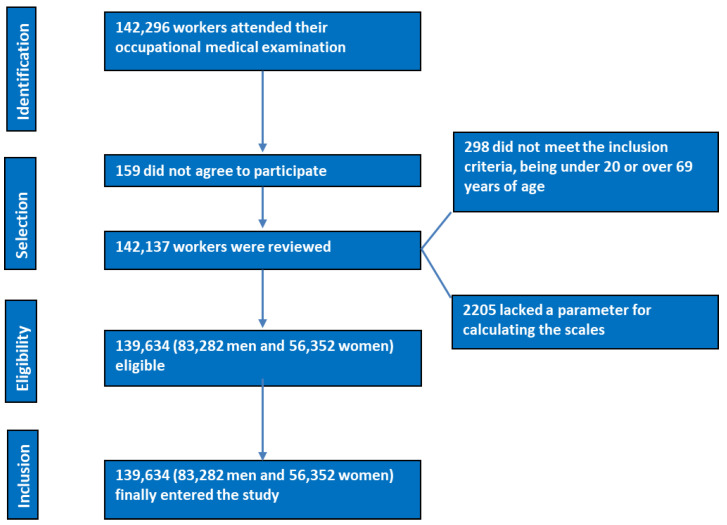
PRISMA flowchart of participants in the study.

**Figure 2 diagnostics-14-02388-f002:**
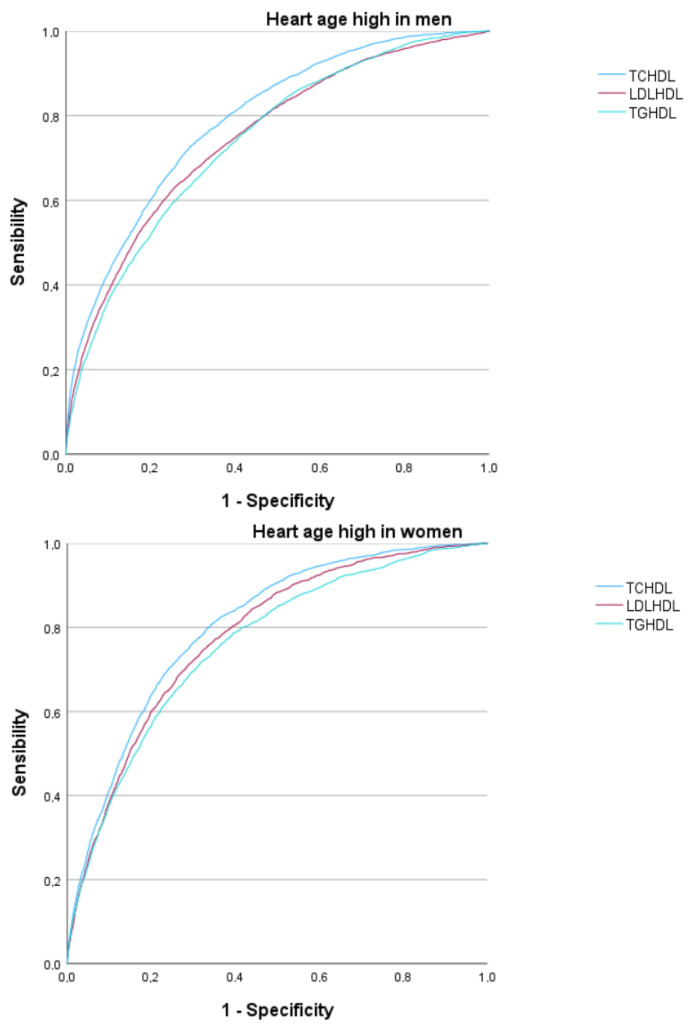
ROC curve.

**Table 1 diagnostics-14-02388-t001:** Variables and scores used to calculate cardiac age.

**Men**											
**Total Cholesterol (mg/dL)**	**Points**	**HDL-c (md/dL)**	**Points**	**Glucose (mg/dL)**	**Points**	**Smokers**	**Points**	**BMI (kg/m^2^)**	**Points**	**SBP/DBP mm Hg**	**Points**
<130	−5	<35	2	<70	−5	Yes	5	<20	−2	<120/<80	−3
130–145	−4	35–39	1	70–80	−3	Non	0	20–22.5	−1	120–139/80–89	0
146–160	−3	40–45	0	81–90	−2			22.6–24.9	0	140–159/90–99	2
161–175	−2	46–50	−1	91–99	0			25–29.9	1	≥160/≥100 *	3
176–190	−1	51–55	−2	100–109	2			30–34.9	2		
191–199	0	56–60	−3	110–125	3			≥35	3		
200–239	1	61–65	−4	>125 *	5						
>240	2	>65	−5								
**Women**											
**Total Cholesterol (mg/dL)**	**Points**	**HDL-c (md/dL)**	**Points**	**Glucose (mg/dL)**	**Points**	**Smokers**	**Points**	**BMI (kg/m^2^)**	**Points**	**SBP/DBP mm Hg**	**Points**
<130	−5	<40	2	<70	−5	Yes	5	<20	−2	<120/<80	−3
130–145	−4	40–49	1	70–80	−3	Non	0	20–22.5	−1	120–139/80–89	0
146–160	−3	50–55	0	81–90	−2			22.6–24.9	0	140–159/90–99	2
161–175	−2	56–60	−1	91–99	0			25–29.9	1	≥160/≥100 *	3
176–190	−1	61–65	−2	100–109	2			30–34.9	2		
191–199	0	66–70	−3	110–125	3			≥35	3		
200–239	1	71–75	−4	>125 *	5						
>240	2	<75	−5								

(*) Or in treatment.

**Table 2 diagnostics-14-02388-t002:** Characteristics of the population.

	Men *n* = 83,282	Women *n* = 56,352	All *n* = 139,634	
	Mean (SD)	Mean (SD)		*p*-Value
**Age (years)**	41.4 (10.7)	40.1 (10.4)	40.9 (10.6)	<0.0001
**Height (cm)**	173.8 (7.1)	161.2 (6.5)	168.7 (9.2)	<0.0001
**Weight (kg)**	83.2 (14.6)	66.3 (13.9)	76.3 (16.5)	<0.0001
**Body mass index (kg/m^2^)**	27.5 (4.5)	25.5 (5.3)	26.7 (4.9)	<0.0001
**Waist circumference (cm)**	90.2 (10.3)	76.3 (10.5)	84.6 (12.4)	<0.0001
**Waist-to-height ratio**	0.52 (0.06)	0.47 (0.07)	0.50 (0.07)	<0.0001
**Systolic blood pressure (mmHg)**	126.2 (15.9)	115.6 (15.7)	121.9 (16.7)	<0.0001
**Diastolic blood pressure (mmHg)**	76.6 (10.9)	71.1 (10.7)	74.4 (11.1)	<0.0001
**Total cholesterol (mg/dL)**	199.6 (38.6)	194.6 (36.9)	197.6 (38.0)	<0.0001
**HDL/cholesterol (mg/dL)**	50.0 (7.7)	54.7 (9.2)	51.9 (8.6)	<0.0001
**LDL/cholesterol (mg/dL)**	122.6 (37.4)	121.5 (37.1)	122.2 (37.3)	<0.0001
**Triglycerides (mg/dL)**	133.8 (95.6)	90.8 (49.7)	116.4 (83.0)	<0.0001
**Glycaemia (mg/dL)**	93.0 (25.4)	86.8 (18.1)	90.5 (22.9)	<0.0001
	***n* (%)**	***n* (%)**		***p*-Value**
**20–29 years**	12,558 (15.1)	10,110 (18.0)	22,668 (16.2)	<0.0001
**30–39 years**	24,648 (29.6)	17,460 (31.0)	42,108 (30.2)	
**40–49 years**	25,178 (30.2)	17,094 (30.3)	42,272 (30.3)	
**50–59 years**	17,370 (20.9)	9984 (17.7)	27,354 (19.6)	
**60–70 years**	3528 (4.2)	1704 (3.0)	5232 (3.7)	
**Social class I**	6234 (7.5)	7632 (13.6)	13,866 (9.9)	<0.0001
**Social class II**	19,856 (23.8)	18,112 (32.1)	37,968 (27.2)	
**Social class III**	57,192 (68.7)	30,608 (54.3)	87,800 (62.9)	
**Elementary school**	55,306 (66.4)	27,086 (48.1)	82,392 (59.0)	
**High school**	22,408 (26.9)	22,574 (40.0)	44,982 (32.2)	
**University**	5568 (6.7)	6692 (11.9)	12,260 (8.8)	
**Non-smokers**	55,618 (66.8)	38,252 (67.9)	93,870 (67.2)	<0.0001
**Smokers**	27,664 (33.2)	18,100 (32.1)	45,764 (32.8)	
**No physical activity**	51,984 (62.4)	28,962 (51.4)	80,946 (58.0)	<0.0001
**Yes physical activity**	31,298 (37.6)	27,390 (48.6)	58,688 (42.0)	
**No Mediterranean diet**	54,792 (65.8)	29,764 (52.8)	84,556 (60.6)	<0.0001
**Yes Mediterranean diet**	28,490 (34.2)	26,588 (47.2)	55,078 (39.4)	
**No alcohol consumption**	56,022 (67.3)	47,536 (84.4)	103,558 (74.2)	<0.0001
**Yes alcohol consumption**	27,260 (32.7)	8816 (15.6)	36,076 (25.8)	

HDL—high-density lipoprotein. LDL—low-density lipoprotein.

**Table 3 diagnostics-14-02388-t003:** Mean values of ALLY heart age according to levels of different atherogenic indices by sex.

		Men			Women	
ALLY HA	*n*	Mean (SD)	*p*-Value	*n*	Mean (SD)	*p*-Value
**Low TC/HDL-c**	6,8020	5.2 (7.6)	<0.001	45,174	−0.4 (8.8)	<0.001
**Moderate TC/HDL-c**	14,988	13.7 (6.3)		10,730	9.7 (8.6)	
**High TC/HDL-c**	274	18.8 (3.0)		448	15.2 (7.3)	
**Normal LDL-c/HDL-c**	70,700	5.5 (7.7)	<0.001	42,996	−0.6 (8.8)	<0.001
**High LDL-c/HDL-c**	12,582	13.7 (6.3)		13,356	9.0 (8.8)	
**Normal Triglycerides/HDL-c**	59,156	4.9 (7.6)	<0.001	51,302	0.8 (9.3)	<0.001
**High Triglycerides/HDL-c**	24,126	11.3 (7.5)		5050	10.6 (8.8)	

HDL—high-density lipoprotein. LDL—low-density lipoprotein. TC—total cholesterol. ALLY HA—avoidable lost life years heart age. SD—standard deviation.

**Table 4 diagnostics-14-02388-t004:** The influence of age on the relationship between atherogenic indices and heart age.

		**Total Cholesterol/HDL-c ***	**LDL-c/HDL-c ***	**TG/HDL-c ***	**ALLY HA ***
**Men**	** *n* **	**Mean (SD)**	**Mean (SD)**	**Mean (SD)**	**Mean (SD)**
**20–29 years**	12,558	3.3 (0.9)	1.9 (0.8)	1.9 (1.4)	1.3 (4.9)
**30–39 years**	24,648	3.9 (1.1)	2.4 (0.9)	2.5 (2.3)	4.2 (6.7)
**40–49 years**	25,178	4.3 (1.2)	2.7 (1.0)	3.1 (2.8)	7.9 (8.1)
**50–59 years**	17,370	4.6 (1.2)	2.9 (1.0)	3.4 (2.8)	10.7 (7.9)
**60–70 years**	3528	4.7 (1.2)	3.0 (1.1)	3.5 (2.2)	11.8 (7.4)
		**Total Cholesterol/HDL-c ***	**LDL-c/HDL-c ***	**TG/HDL-c ***	**ALLY HA ***
**Women**	** *n* **	**Mean (SD)**	**Mean (SD)**	**Mean (SD)**	**Mean (SD)**
**20–29 years**	10,110	3.1 (0.9)	1.8 (0.8)	1.4 (0.8)	−2.1 (5.0)
**30–39 years**	17,640	3.4 (1.0)	2.1 (0.9)	1.6 (0.9)	−1.8 (7.7)
**40–49 years**	17,094	3.8 (1.0)	2.5 (0.9)	1.8 (1.2)	2.8 (10.2)
**50–59 years**	9984	4.3 (1.1)	2.8 (1.0)	2.1 (1.4)	8.4 (10.6)
**60–70 years**	1704	4.4 (1.1)	2.9 (1.0)	2.3 (1.2)	8.6 (9.9)

(*) Statistical significance (*p* < 0.001) HDL-c—high-density lipoprotein–cholesterol. LDL-c—low-density lipoprotein–cholesterol. TG—triglycerides. ALLY HA—avoidable lost life years heart age. SD—standard deviation.

**Table 5 diagnostics-14-02388-t005:** Prevalence of high values of heart age according to levels of different atherogenic indices by sex.

		Men			Women	
High ALLY HA	*n*	%	*p*-Value	*n*	%	*p*-Value
**Low TC/HDL-c**	68,020	11.6	<0.001	45,174	7.0	<0.001
**Moderate TC/HDL-c**	14,988	44.9		10,730	32.0	
**High TC/HDL-c**	274	86.1		448	63.8	
**Normal LDL-c/HDL-c**	70,700	12.9	<0.001	42,996	7.6	<0.001
**High LDL-c/HDL-c**	12,582	45.4		13,356	29.7	
**Normal Triglycerides/HDL-c**	59,156	10.9	<0.001	51,302	9.7	<0.001
**High Triglycerides/HDL-c**	24,126	70.1		5050	37.7	

HDL—high-density lipoprotein. LDL—low-density lipoprotein. TC—total cholesterol. ALLY—avoidable lost life years heart age.

**Table 6 diagnostics-14-02388-t006:** Multinomial logistic regression.

	20–29 Years	30–39 Years	40–49 Years	50–59 Years	60–70 Years
High ALLY Heart Age	OR (95% CI)	OR (95% CI)	OR (95% CI)	OR (95% CI)	OR (95% CI)
**Women**	1	1	1	1	1
**Men**	1.25 (1.20–1.31)	1.67 (1.46–1.88)	1.54 (1.43–1.65)	1.12 (1.05–0.20)	1.32 (1.24–1.40)
**Social class I**	1	1	1	1	1
**Social class II**	2.01 (1.86–2.16)	1.96 (1.63–2.30)	1.88 (1.71–2.05)	1.87 (1.71–2.04)	1.97 (1.69–2.25)
**Social class III**	2.41 (2.20–2.62)	3.30 (2.67–3.94)	2.37 (2.00–2.75)	4.20 (3.72–4.69)	5.21 (4.70–5.72)
**University**	1	1	1	1	1
**High school**	1.85 (1.69–2.02)	1.49 (1.29–1.70)	1.38 (1.29–1.48)	1.52 (1.38–1.66)	1.48 (1.30–1.67)
**Elementary school**	2.58 (2.30–2.67)	2.62 (2.30–2.95)	2.68 (2.40–2.97)	2.46 (2.09–2.82)	2.40 (2.11–2.70)
**Non-smokers**	1	1	1	1	1
**Smokers**	29.15 (25.45–32.85)	22.33 (21.0–23.67)	21.50 (19.93–23.08)	13.23 (12.28–14.19)	4.97 (4.23–5.70)
**Physical activity**	1	1	1	1	1
**No physical activity**	1.63 (1.50–1.76)	3.32 (2.68–3.96)	4.15 (3.70–4.60)	4.45 (3.65–5.26)	5.98 (5.01–6.95)
**Mediterranean diet**	1	1	1	1	1
**No Mediterranean diet**	1.34 (1.25–1.44)	1.89 (1.50–2.29)	2.27 (2.01–2.53)	2.39 (2.09–2.70)	3.11 (2.60–3.62)
**No alcohol consumption**	1	1	1	1	1
**Alcohol consumption**	3.43 (2.98–3.88)	4.04 (3.51–4.58)	4.33 (3.90–4.77)	7.34 (6.04–8.65)	7.45 (6.02–7.88)
**TC/HDL-c low**	1	1	1	1	1
**TC/HDL-c moderate**	7.13 (6.60–7.67)	6.45 (5.01–7.90)	5.44 (5.07–5.82)	4.02 (3.60–4.44)	3.00 (2.60–3.41)
**TC/HDL-c high**	29.98 (27.02–32.94)	20.24 (18.01–22.47)	16.52 (14.99–18.05)	7.34 (6.00–8.68)	6.90 (6.02–7.79)
**LDL-c/HDL-c low**	1	1	1	1	1
**LDL-c/HDL-c high**	2.99 (2.51–3.47)	2.61 (2.37–2.85)	2.46 (2.11–2.82)	2.29 (2.02–2.57)	2.22 (1.80–2.64)
**Triglycerides/HDL-c low**	1	1	1	1	1
**Triglycerides/HDL-c high**	2.45 (2.20–2.30)	1.90 (1.70–2.11)	1.72 (1.49–1.95)	1.54 (1.41–1.67)	1.41 (1.31–1.52)

HDL—high-density lipoprotein. LDL—low-density lipoprotein. TC—total cholesterol. ALLY—avoidable lost life years. OR—odds ratio. In all cases, the differences are statistically significant.

**Table 7 diagnostics-14-02388-t007:** Area under the curve by sex.

	Men	Women
	AUC (95% CI)	AUC (95% CI)
**Total cholesterol/HDL-c**	0.790 (0.786–0.794)	0.803 (0.798–0.808)
**LDL-c/HDL-c**	0.750 (0.746–0.754)	0.780 (0.774–0.785)
**Triglycerides/HDL-c**	0.742 (0.737–0.746)	0.760 (0.754–0.766)

HDL—high-density lipoprotein. LDL—low-density lipoprotein. AUC—area under the curve.

**Table 8 diagnostics-14-02388-t008:** Cut-off for atherogenic indices to predict high values of heart age.

	Men	Women
	Cut-Off–Sensitivity–Specificity–Youden	Cut-Off–Sensitivity–Specificity–Youden
**Total cholesterol/HDL-c**	4.30–71.6–71.5–0.431	4.00–73.8–72.4–0.464
**LDL-c/HDL-c**	2.71–68.2–68.2–0.364	2.60–71.2–71.0–0.422
**Triglycerides/HDL-c**	2.53–67.0–67.0–0.340	1.73–70.0–69.5–0.395

HDL—high-density lipoprotein. LDL—low-density lipoprotein.

## Data Availability

This study’s data are stored in a database that complies with all security measures at ADEMA-Escuela Universitaria. The Data Protection Delegate is Ángel Arturo López González.

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
