# Peer review of "Usefulness of Atherogenic Indices for Predicting High Values of Avoidable Lost Life Years Heart Age in 139,634 Spanish Workers"

_diagnostics, 2024, doi:10.3390/diagnostics14212388_

Round 1

Reviewer 1 Report

Comments and Suggestions for Authors

Dear Authors,

Firstly, I would like to express my gratitude for the opportunity to review your manuscript titled “Usefulness of Atherogenic Indices for Predicting High Values of Ally Heart Age in 139,634 Spanish Workers” The study presents interesting data on the relationship between atherogenic indices and heart age among Spanish workers. However, I have several major concerns that I hope you will address.

Major:

  1. The introduction does not clearly articulate the primary significance of the study. Since heart age is calculated using a Heart Age Calculator, the novelty or importance of correlating it with atherogenic indices may be limited. I suggest that the authors more explicitly describe the innovation and potential scientific or clinical contributions of this study in the introduction.
  2. The discussion section, in some parts, reads more like a review than an analysis of the study’s findings. The authors should summarize the main findings and provide an analysis of their implications, including potential mechanisms and clinical applications. Additionally, it would be beneficial to discuss how the results of this study compare and contrast with other relevant research, including possible reasons for any differences.
  3. Given the study’s focus on heart age, it would be valuable to include subgroup analyses by different age groups to explore the influence of age on the relationship between atherogenic indices and heart age. This would further elucidate the clinical relevance of these indices at various stages of life.

Minor :

  1. The use of abbreviations, particularly for Heart Age, appears to be inconsistent throughout the manuscript. After the first mention of the full term, the authors should maintain consistent use of abbreviations throughout the paper to avoid confusion.

I believe that with these revisions, the manuscript will significantly improve and provide a clearer contribution to the existing body of research on heart age and atherogenic indices.

Comments on the Quality of English Language

none

Author Response

Reviewer 1

Dear reviewer,

First of all, thank you for your work and all your recommendations.

To facilitate your review, we have written the modifications in red in the article.

Major:

  1. The introduction does not clearly articulate the primary significance of the study. Since heart age is calculated using a Heart Age Calculator, the novelty or importance of correlating it with atherogenic indices may be limited. I suggest that the authors more explicitly describe the innovation and potential scientific or clinical contributions of this study in the introduction.

Thank you for your recommendation. Given that cardiovascular diseases are the leading cause of morbidity and mortality worldwide, and considering that, according to the WHO, preventive measures are the most effective way to prevent them, we sought a more comprehensible element for patients that could serve as a motivational tool to modify their harmful lifestyle habits.

If we find an association between atherogenic indices and heart age, we believe that one of the simplest ways to motivate patients to change their harmful behaviors is by making them aware of their potential years of life lost (YPLL). The clinical utility of this study lies in the fact that by calculating a patient’s atherogenic index during consultation, we can highlight their potential years of life lost, thereby raising awareness of the need for lifestyle changes.

We have added a paragraph to the introduction, justifying the scientific importance and innovation of the study.

  1. The discussion section, in some parts, reads more like a review than an analysis of the study’s findings. The authors should summarize the main findings and provide an analysis of their implications, including potential mechanisms and clinical applications. Additionally, it would be beneficial to discuss how the results of this study compare and contrast with other relevant research, including possible reasons for any differences.

We sincerely appreciate your advice once again. We have revised the discussion and compared our study with other publications that are in some way related to our objective. However, it has not been possible to compare it with studies that conduct the same type of analysis, as our literature search did not yield any publications referring to the relationship between atherogenic indices and Heart Age.

  1. Given the study’s focus on heart age, it would be valuable to include subgroup analyses by different age groups to explore the influence of age on the relationship between atherogenic indices and heart age. This would further elucidate the clinical relevance of these indices at various stages of life.

We completely agree with you, and we have proceeded to include a subgroup analysis by different age groups to explore the influence of age in relation to atherogenic indices and heart age (Table 3). We believe this enhances the scientific interest of the paper, clarifies the results, and improves the overall quality. Thank you very much.

Minor :

  1. The use of abbreviations, particularly for Heart Age, appears to be inconsistent throughout the manuscript. After the first mention of the full term, the authors should maintain consistent use of abbreviations throughout the paper to avoid confusion.

You are absolutely right, thank you. Following your recommendation, we have standardized the use of abbreviations throughout the manuscript.

I believe that with these revisions, the manuscript will significantly improve and provide a clearer contribution to the existing body of research on heart age and atherogenic indices.

Thank you very much for your suggestions. We have proceeded to answer all of them and we trust that they will adequately respond to your questions.

Reviewer 2 Report

Comments and Suggestions for Authors

The authors present an original study entitled “USEFULNESS OF ATHEROGENIC INDICES FOR PREDICTING HIGH VALUES OF ALLY HEART AGE IN 139,634 SPANISH WORKERS”.

The results of the study are quite interesting and the large sample size is definitely a strength of the study.

However, I have a few major points to address:

1.    If “ALLY” is an abbreviation, it either needs to be written in capital letters or in full in the title of the article.

2.    “By using the Friedewald formula, which is only reliable when triglycerides do not exceed 400, LDL-cholesterol can be calculated indirectly”. Please add a unit of measurement after the number 400. Also, please add a reference supporting this statement.

3.    “Atherogenic index triglycerides/HDL-c”. Please add the ‘=’ symbol for uniformity in equations descriptions. Also please provide references to the sources of the relevant equations.

It should be noted that the formula for calculating AIP used by the authors differs from the formula most often reported in other studies, for example:

-        M. Dobiášová. Atherogenic Index of Plasma [Log(Triglycerides/HDL-Cholesterol)]: Theoretical and Practical Implications. Clinical Chemistry. 2004; 50(7):1113-1115.

-        Li YW, Kao TW, Chang PK, Chen WL, Wu LW. Atherogenic index of plasma as predictors for metabolic syndrome, hypertension and diabetes mellitus in Taiwan citizens: a 9-year longitudinal study. Sci Rep. 2021 May 10;11(1):9900.

-        Kammar-García A, López-Moreno P, Hernández-Hernández ME, Ortíz-Bueno AM, Martínez-Montaño MLC. Atherogenic index of plasma as a marker of cardiovascular risk factors in Mexicans aged 18 to 22 years. Proc (Bayl Univ Med Cent). 2020 Aug 21;34(1):22-27.

-        Kim SH, Cho YK, Kim YJ, et al. Association of the atherogenic index of plasma with cardiovascular risk beyond the traditional risk factors: a nationwide population-based cohort study. Cardiovasc Diabetol. 2022;21(1):81. 

In addition, several sources state that TG and HDL values should be expressed in mmol/L.

Hence, the selected cut-off values for AIP also require clarification.

Authors need to clarify whether these are different AIPs, and if these are different AIPs, they need to provide a reference to the source of their equation and explain its choice.

4.    “A person was considered a smoker if they had quit smoking less than a year before or if they had smoked one cigarette per day for the preceding thirty days”. That's an interesting definition. Is that the original author's definition? I'm sure the authors realise that the results of their study should be presented in a way that allows comparison with the results of other studies. It is important that definitions of certain conditions are generally accepted. Please, see Klemperer EM, Hughes JR, Callas PW, West JC, Villanti AC. Tobacco and Nicotine Use Among US Adult "Never Smokers" in Wave 4 (2016-2018) of the Population Assessment of Tobacco and Health Study. Nicotine Tob Res. 2021 Jun 8;23(7):1199-1207.

5.    Please add one more column to Table 1 with data for the total sample (males and females).

6.    The authors report all lipid data as mean and standard deviation. Were all these variables characterised by a normal distribution?

7.    Explanations are required to be added to the figures. Here appears probably a reference to the formula ‘logTGHDL’, which was not there before. This is extremely confusing.

8.    “By using the Friedewald formula, which is only reliable when triglycerides do not exceed 400, LDL-cholesterol can be calculated indirectly”. Were there any patients with hypertriglyceridaemia in the study? Obviously, some number of them would have been included. How were LDL, and the corresponding indices, determined in this case?

Author Response

Reviewer 2

Dear reviewer,

First of all, thank you for your work and all your recommendations.

To facilitate your review, we have written the modifications in red in the article.

The authors present an original study entitled “USEFULNESS OF ATHEROGENIC INDICES FOR PREDICTING HIGH VALUES OF ALLY HEART AGE IN 139,634 SPANISH WORKERS”.

The results of the study are quite interesting and the large sample size is definitely a strength of the study.

However, I have a few major points to address:

  1. If “ALLY” is an abbreviation, it either needs to be written in capital letters or in full in the title of the article.

Thank you for your observation. We have proceeded to include ALLY in the title of the article.

  1. “By using the Friedewald formula, which is only reliable when triglycerides do not exceed 400, LDL-cholesterol can be calculated indirectly”. Please add a unit of measurement after the number 400. Also, please add a reference supporting this statement.

Following your suggestion, we have added a unit of measurement and the corresponding bibliographic reference. Thank you very much.

Rosales-Rimache J, Apaza-Condori J, Rabanal-Sanchez J, Jari L, Soncco-Llulluy F. Comparison of the Friedewald and Vujovic methods with the calculated LDL concentration in a biochemical auto-analyzer. Medwave. 2024 May 6;24(4):e2775. doi: 10.5867/medwave.2024.04.2775.

  1. “Atherogenic index triglycerides/HDL-c”. Please add the ‘=’ symbol for uniformity in equations descriptions. Also please provide references to the sources of the relevant equations.

We have included the "=" symbol.

It should be noted that the formula for calculating AIP used by the authors differs from the formula most often reported in other studies, for example:

-        M. Dobiášová. Atherogenic Index of Plasma [Log(Triglycerides/HDL-Cholesterol)]: Theoretical and Practical Implications. Clinical Chemistry. 2004; 50(7):1113-1115.

-        Li YW, Kao TW, Chang PK, Chen WL, Wu LW. Atherogenic index of plasma as predictors for metabolic syndrome, hypertension and diabetes mellitus in Taiwan citizens: a 9-year longitudinal study. Sci Rep. 2021 May 10;11(1):9900.

-        Kammar-García A, López-Moreno P, Hernández-Hernández ME, Ortíz-Bueno AM, Martínez-Montaño MLC. Atherogenic index of plasma as a marker of cardiovascular risk factors in Mexicans aged 18 to 22 years. Proc (Bayl Univ Med Cent). 2020 Aug 21;34(1):22-27.

-        Kim SH, Cho YK, Kim YJ, et al. Association of the atherogenic index of plasma with cardiovascular risk beyond the traditional risk factors: a nationwide population-based cohort study. Cardiovasc Diabetol. 2022;21(1):81. 

In addition, several sources state that TG and HDL values should be expressed in mmol/L.

The reference laboratories provided the results in mg/dL rather than mmol/L. Although converting mg/dL to mmol/L only requires dividing by 18, this operation would need to be performed on 139,634 analytical results, followed by redoing the entire statistical analysis. We kindly request that you accept the results in mg/dL, which are the standard reference units used by all laboratories in Spain. Thank you very much.

Hence, the selected cut-off values for AIP also require clarification.

The triglycerides/HDL-C ratio is a widely used index in Spain and differs from the logarithmic triglycerides/HDL-C ratio. To determine the cut-off points for atherogenic indices, we relied on the recommendations from the guidelines established by the Spanish Ministry of Health.

Authors need to clarify whether these are different AIPs, and if these are different AIPs, they need to provide a reference to the source of their equation and explain its choice.

Yes, it is a different AIP. We based our work on the recommendations from the guidelines established by the Spanish Ministry of Health. We have added the reference.

Morales MT, Hijano-Villegas S, Martínez-Llamas MS, López-Barba J, Díaz-Portillo J. Guía del paciente con trastornos lipídicos. Ministerio de Sanidad y Consumo. Instituto Nacional de Gestión Sanitaria. 2007

  1. “A person was considered a smoker if they had quit smoking less than a year before or if they had smoked one cigarette per day for the preceding thirty days”. That's an interesting definition. Is that the original author's definition? I'm sure the authors realise that the results of their study should be presented in a way that allows comparison with the results of other studies. It is important that definitions of certain conditions are generally accepted. Please, see Klemperer EM, Hughes JR, Callas PW, West JC, Villanti AC. Tobacco and Nicotine Use Among US Adult "Never Smokers" in Wave 4 (2016-2018) of the Population Assessment of Tobacco and Health Study. Nicotine Tob Res. 2021 Jun 8;23(7):1199-1207.

We apologize for the mistake and have made the correction. It was an error in the phrasing of the sentence.

“Anyone who had smoked at least one cigarette in the previous thirty days (or its equivalent in other forms of consumption) or who had stopped smoking less than a year before was considered a smoker”.

  1. Please add one more column to Table 1 with data for the total sample (males and females).

Following your instructions, we have added a column to Table 1 with data for the entire sample (both men and women).

  1. The authors report all lipid data as mean and standard deviation. Were all these variables characterised by a normal distribution?

      Yes, all variables were normally distributed, so the mean and standard deviation were calculated.

  1. Explanations are required to be added to the figures. Here appears probably a reference to the formula ‘logTGHDL’, which was not there before. This is extremely confusing.

By mistake, the ROC curves included the logarithm of triglycerides/HDL-c instead of triglycerides/HDL-c. We have corrected this. Thank you very much for your observation.

  1. “By using the Friedewald formula, which is only reliable when triglycerides do not exceed 400, LDL-cholesterol can be calculated indirectly”. Were there any patients with hypertriglyceridaemia in the study? Obviously, some number of them would have been included. How were LDL, and the corresponding indices, determined in this case?

When triglyceride levels exceed 400 mg/dL, the Friedewald formula cannot be applied, and the laboratory will perform a direct determination of LDL-c, providing us with the results.

Thank you very much for your suggestions. We have proceeded to answer all of them and we trust that they will adequately respond to your questions.

Round 2

Reviewer 1 Report

Comments and Suggestions for Authors

Thank you for inviting me to review the manuscript again.

After reviewing the manuscript, I noticed that the authors failed to follow my suggestion of conducting age-stratified subgroup analyses within the Multinomial logistic regression framework for the primary objective of assessing the "Usefulness of Atherogenic Indices for Predicting High Values of ALLY Heart Age." This analysis is crucial for gaining deeper insights into the relationship between cardiovascular risk factors and ALLY heart age across different age groups, thereby enhancing the study's comprehensiveness and the applicability of its conclusions. 

Furthermore, it is imperative to clearly state the confounding variables that were adjusted for in the analysis, such as gender and education level, as these can significantly influence the associations being examined.

The current discussion also lacks focus on the main theme of the study, i.e., the association between atherogenic indices and heart age. For instance, the inclusion of discussions on excessive alcohol consumption's impact on cardiovascular health seems tangential to the research question at hand, as the study's primary aim does not investigate cardiovascular risk factors broadly but specifically focuses on the relationship between atherogenic indices and heart age. I recommend the authors to streamline their discussion, focusing solely on how the findings relate to and support the study's central hypothesis.

Comments on the Quality of English Language

none

Author Response

Reviewer 1

Dear reviewer,

First of all, thank you for your work and all your recommendations.

To facilitate your review, we have written the modifications in red in the article.

Major:

After reviewing the manuscript, I noticed that the authors failed to follow my suggestion of conducting age-stratified subgroup analyses within the Multinomial logistic regression framework for the primary objective of assessing the "Usefulness of Atherogenic Indices for Predicting High Values of ALLY Heart Age." This analysis is crucial for gaining deeper insights into the relationship between cardiovascular risk factors and ALLY heart age across different age groups, thereby enhancing the study's comprehensiveness and the applicability of its conclusions. 

We apologize for not adequately addressing your suggestion in the first review. Please accept our apologies, as we did not fully understand what you were requesting. We have performed the multinomial logistic regression that we believe you were asking for.

Furthermore, it is imperative to clearly state the confounding variables that were adjusted for in the analysis, such as gender and education level, as these can significantly influence the associations being examined.

We have included the various confounding variables in the analysis to avoid potential biases associated with them.

The current discussion also lacks focus on the main theme of the study, i.e., the association between atherogenic indices and heart age. For instance, the inclusion of discussions on excessive alcohol consumption's impact on cardiovascular health seems tangential to the research question at hand, as the study's primary aim does not investigate cardiovascular risk factors broadly but specifically focuses on the relationship between atherogenic indices and heart age. I recommend the authors to streamline their discussion, focusing solely on how the findings relate to and support the study's central hypothesis.

Following your suggestion, we have also revised the discussion to establish the relationship between each variable and its influence on heart age.

Thank you very much for your suggestions. We have proceeded to answer all of them and we trust that they will adequately respond to your questions.